# 3D Printed Surgical Guide for Coronary Artery Bypass Graft: Workflow from Computed Tomography to Prototype

**DOI:** 10.3390/bioengineering9050179

**Published:** 2022-04-19

**Authors:** Ida Anna Cappello, Mara Candelari, Luigi Pannone, Cinzia Monaco, Edoardo Bori, Giacomo Talevi, Robbert Ramak, Mark La Meir, Ali Gharaviri, Gian Battista Chierchia, Bernardo Innocenti, Carlo de Asmundis

**Affiliations:** 1Heart Rhythm Management Centre, Postgraduate Program in Cardiac Electrophysiology and Pacing, Universitair Ziekenhuis Brussel Vrije Universiteit Brussel, European Reference Networks Guard-Heart, 1090 Brussels, Belgium; ida.cappello@gmail.com (I.A.C.); mara.candelari@gmail.com (M.C.); lui.pannone@gmail.com (L.P.); cinziamonaco90@gmail.com (C.M.); talev1995@gmail.com (G.T.); robbert.ramak@uzbrussel.be (R.R.); ali.gharaviri@gmail.com (A.G.); gbchier@yahoo.it (G.B.C.); 2BEAMS Department, Bio Electro and Mechanical Systems, École Polytechnique de Bruxelles, Université Libre de Bruxelles, 1050 Brussels, Belgium; edoardo.bori@gmail.com (E.B.); bernardo.innocenti@ulb.be (B.I.); 3Cardiac Surgery Department, Universitair Ziekenhuis Brussel—Vrije Universiteit Brussel, 1090 Brussels, Belgium; lameir@yahoo.com

**Keywords:** 3D printing, pre-operative planning, image processing, segmentation

## Abstract

Patient-specific three-dimensional (3D) printed models have been increasingly used in many medical fields, including cardiac surgery for which they are used as planning and communication tools. To locate and plan the correct region of interest for the bypass placement during coronary artery bypass graft (CABG) surgery, cardiac surgeons can pre-operatively rely on different medical images. This article aims to present a workflow for the production of a patient-specific 3D-printed surgical guide, from data acquisition and image segmentation to final prototyping. The aim of this surgical guide is to help visualize the region of interest for bypass placement during the operation, through the use of dedicated surgical holes. The results showed the feasibility of this surgical guide in terms of design and fitting to the phantom. Further studies are needed to assess material biocompatibility and technical properties.

## 1. Introduction

Over the years, different tools have been developed to visualize coronary artery stenosis and for the planning of coronary artery bypass graft (CABG) surgery. These tools consistently rely on medical images, mainly acquired via Computed Tomography (CT) and Magnetic Resonance Imaging (MRI) [1]. However, the 2D clinical images used for surgical planning do not provide a detailed visualization of the cardiovascular anatomy. Furthermore, 3D clinical image reconstruction does not allow for the assessment of pathological areas located directly on the heart.

Therefore, a 3D-printed model, using image segmentation, may be very helpful to compensate for bidimensional image deficiencies while also paving the way towards a broad range of innovative applications in the field [2].

In particular, the critical steps in CABG surgery are as follows: (1) To locate the site of coronary artery narrowing, given the fact that the heart can be covered by fat which can prevent the cardiac surgeon from obtaining an optimal view; (2) To find the best anatomical site, after a critical stenosis, for the bypass placement. These issues cannot be overcome with conventional clinical images during surgery.

A 3D-printed coronary artery tree map model, reconstructed by manipulating 2D images obtained from the CT, can be used in order to address these issues. The 3D model can be employed as both a surgical educational tool and to help cardiac surgeons in visualizing and planning complex CABG surgery [3].

The aim of this research is to describe in detail the workflow for the creation of a patient-specific 3D-printed tool to be used as a surgical guide during CABG surgery (Figure 1). Furthermore, a fitting test of the surgical guide on the 3D-printed heart phantom was aimed.

## 2. Materials and Methods

Data acquisition and image segmentation represent the essential processes for analyzing the anatomical structure of interest, consisting in obtaining a series of 2D images, and then employing them for the 3D model reconstruction [4]. The 3D model generated from the segmentation is still not suitable for printing. An accurate phase that involves one or more CADs software is necessary to obtain the final printed 3D model.

A workflow for the 3D surgical guide is hereby described. It consists of sequential steps starting from image segmentation of a heart model using 2D images, ending with the actual surgical guide production via 3D printing.

### 2.1. Data Acquisition and Images Segmentation: 3D Heart Model Design

The data acquisition consisted in a collection of cardiovascular CT images deemed useful for reconstructing, through image segmentation, a 3D model of the anatomy of the whole heart with coronary artery tree, in order to create a 3D heart template for the building of the surgical guide. A 256-Slice CT Scanner (GE Healthcare system) was used for data acquisition of heart and intravascular structures, obtaining a good quality of images for 3D reconstruction. Moreover, the quality of the 3D models depends on the properties of the medical imaging data used. In fact, accurate models can be created when the slice thickness of the images are 0.50–1.25 mm; however, the optimal size depends on the pathology of interest [5]. In this study, a 256-slice CT scanner was used, with a slice thickness of 0.625 mm and a size of 512 × 512 pixels, to obtain a faithful reconstruction of the heart and coronary arteries. The medical images were stored in Digital Imaging and Communications in Medicine (DICOM) format, which is the standard for storing and transmitting patient data.

Following data acquisition and clinical consideration that confirmed the presence of stenosis to be treated by CABG surgery in different coronary arteries, the second step involved the reconstruction of the 3D heart model based on image segmentation. The DICOM CT images were transferred and manipulated in 3D Slicer (Brigham and Women’s Hospital, Boston, MA, USA).

A semi-automated approach, generating a 3D volume from all areas of the images that respected the threshold range imposed, was used. To ensure quality, subsequent refining was performed manually slice-by-slice.

Three stages of image segmentation were used for reconstruction of the 3D heart model.

The first stage consisted in whole heart surface reconstruction, without consideration of the coronary artery tree structure. A threshold range between 1.00 and 375 was used to identify the whole epicardial boundary (the green volume shown in Figure 2A). Manual editing and image filtering were then applied to isolate the heart geometries from the surrounding soft tissues, bones or other anatomical structures, and to smoothen the surface of the heart.

For the second stage of image segmentation, a semi-automated approach to reconstruct the first tract of the aorta and coronary arteries was used. The threshold range was set between 99 and 1700, because of the contrast agent used during image acquisition that brightens the shades of gray of the intravascular structures. However, further manipulation and manual editing were performed to mark the coronary tract which followed each stenosis. The optimal region for the operational cut was discussed with the medical staff. Accordingly, for the bypass placement, the regions of interest were defined as the tracts of the coronary arteries from 1 mm ± 0.5 mm after the distal stenosis to the last surgical approachable point. The latter, representing the distal end of the region of interest, was defined as the site where the vessel diameter reached 1.5 mm (Figure 2B). Based on these regions, the operative holes were designed on the 3D surgical guide (Figure 2B).

The third stage of image segmentation was to apply a merging logic operator to obtain the final segmentation, resulting in a combination between the segmentations above (Figure 2C). The final model was converted into a standard tessellation language (STL) file for 3D printing. This file was generated through complex algorithms such as interpolation, which combines the segmented regions of interest into a 3D model.

The generated STL file required additional optimization and refinement before being ready for use as a template-model for the reconstruction of a 3D coronary artery tree surgical guide. Eventual imprecisions in the segmentation process, leading consequently to defects in the obtained 3D model, such as holes and mesh defects, required therefore further post-processing manipulation.

### 2.2. 3D Surgical Prototype Design

Meshmixer (Autodesk Inc., San Rafael, CA, USA) was used as 3D mesh and modelling CAD software, in which STL files were exported to remove segmentation imprecisions. After optimization of the 3D heart model, a virtual prototype of the surgical guide was designed.

After the refinement of the regions of interest and 3D heart model smoothing (Figure 3), manual selection of the regions of interest (surgical holes) was performed in order to create the surgical guide shell. After an extrusion, the surgical guide with the inner surface fitting with the surface of the heart template was obtained.

To ensure a stable placement of the surgical guide on the heart, reference points were chosen. These were considered as points where the guide could be positioned on the organ to ensure its correct position. The aortic ring was considered as a reference point on the top part of the organ, with the apex as a bottom reference point. A series of structure links, with a perforated design, were added to connect the upper and the lower part to the regions of interest.

The first traced reference point was the aortic ring, defined as a surrounding structure which covers the antero-lateral aortic annulus in half circumference, capable of supporting the entire structure of the surgical guide in the upper part. The operative structure of the surgical guide was designed to cover six branches of coronary arteries: right coronary artery (RCA), including a tract of posterior descending artery (PDA), left circumflex (LCX), left anterior descending (LAD), including diagonal 1 (D1) and diagonal 2 (D2).

The perforated interlinks represented the structures that connected the aortic ring to the surgical holes and also connected the distal end of the surgical holes to the apex. The apex represented the second reference point for structure stability; it was designed to cover the heart apex with a concave shape. The surgical holes were designed upon the regions of interest. In particular, assuming a coronary artery width of 2 mm, the hole was calculated with a 2 ± 0.5 mm margin on both sides of the artery, with a total hole width of 6 ± 0.5 mm. The length of the surgical hole was chosen according to the size of the region of interest and customizable based on clinical judgement.

Once the shell of the entire model was created, a thickness was applied using the Select > Edit > Extrude function in Meshmixer. The “Offset” value was set to 2.5 mm and determined the total surgical guide thickness, while the “Direction” type had to be set to “Normal” in order to apply the extrusion uniformly.

Different sections of the surgical guide had an increased diameter to avoid breakage points, such as: (1) the aortic ring with a total thickness of 5.5 mm ± 0.5 mm; (2) the edges of the surgical holes (0.5 mm higher than the structure thickness) and (3) the apex for a total thickness of 3.5 mm ± 0.5 mm with a diameter of 30 mm. Once the surgical guide was defined, smoothing and other functions were applied to create a proper design of the 3D model. For the link region which connects the aortic ring and the apex, a perforated design was chosen, with a series of holes each 2 mm in diameter, at a distance of 10 mm from each other.

The “make solid” operation by Meshmixer, was used to create the surgical guide STL file to be printed, correcting structural errors as shown in Figure 4.

### 2.3. 3D Printing

The 3D heart model was printed to real scale as a phantom to test the surgical guide fitting. The printing process selected to realize the phantom was the fused deposition modeling (FDM), using a Prusa imk3 3D printer, due to its low cost and ease of use [5]. Polylactide (PLA), a rigid material, was used. The phantom, the heart reproduction with a patient-specific region of interest, was only used to test the fit of the surgical guide.

The Stratasys Object260 Connex1 3D printer was then used to realize the surgical guide. The model was printed in single material mode and the printer was set for glossy printing, in order to obtain a more resistant layer [6]. PolyJet was employed for the printing process, representing one of the most recent rapid prototyping processes available on the market. It is a hybrid between selective hardening and drop deposition. The most important component is the printing head: it applies a liquid compound of reactive monomers and oligomers which polymerize in response to ultraviolet light. The application of the material to the tray is granted by the piezo-electric printing head which injects the liquid compound onto the metallic tray. After the injection of each layer, the tray moves downward one layer in thickness, and then the process is repeated with the next layer [7]. The thickness of each layer measured 16 nanometers; therefore, this process can be ranked among the most accurate processes and the device can be considered as one of the fastest devices for rapid prototyping. Thus, the small layer thickness ensured the manufacture of a model with very smooth surface and small details. The material chosen for this research study was a flexible, transparent and biocompatible material from Stratasys: MED625FLX™ (Stratasys Inc., Eden Prairie, MN, USA) [8].

Material, support material and printing mode were defined before starting the printing. When all settings were selected, the printing time was calculated by the software, accounting for the object position on the tray.

After completion of the printing process, the surgical guide was encased in the SUP706 (FullCure705) (Figure 5A), a gel-like support material which had to be carefully removed to expose the delicate printed portions. To remove support material a high-pressure water jet was used: the Objet WaterJet cleaning unit, also marketed by Stratasys (Figure 5B). The cleaning time varies for each printed model, according to its design.

## 3. Results

Once the 3D model was obtained and the post-processing treatment executed, the physical prototype of the surgical guide appeared as a transparent basket-shaped structure with five surgical holes for each coronary artery of interest, as shown the Figure 6; the measured thickness was 2.5 mm, as expected. The model did not exceed the limits of tolerance in height, width, and thickness of the printer, that is 255 mm × 252 mm × 200 mm. In fact, the final surgical guide had a weight of 35 g and occupied a volumetric space of 28,992 mm^3^ (109 mm × 93 mm × 121 mm). After the end of the printing process, cooling of the material was performed to avoid damage of the structure embedded in the support material and to allow its washing. The washing did not cause any damage to the model. The model exhibited no signs of stress and no printing errors. The printing and cleaning of the model was completed in a 10 h cycle.

The two selected reference points, the aortic ring and the apex, respected the increase in thickness along the entire normal direction during the printing, as desired. The aortic ring had a final thickness of 5.5 mm along the entire semicircular structure. The apex had a thickness of 3.5 mm.

The connection between the aortic ring and apex was designed as a perforated link, interrupted in the central section by holes. The surgical holes had a width triple the size of the vessel to be treated, for optimal visualization of the artery.

The planned six regions of interest in the printed model contained the following surgical holes: (1) 33 mm × 6 mm along RCA; (2) 22 mm × 6 mm along the PDA branch; (3) 38 mm × 6 mm along the LAD branch; (4) 47 mm × 6 mm along D2 branch; (5) 24 mm × 6 mm along D1 branch; (6) 63 mm × 6 mm along LCX branch. The dimensions of the regions of interest were deemed suitable for the purpose by the medical staff. Each region of interest opening was designed with edges along each perimeter, for a total thickness of 4 mm.

The printing was performed in high-speed mode with a total MED625FLX consumption of 140 g and FullCure705 consumption of 289 g. The printing time was 7 h, while the cleaning time was 2 h, 1 h less than expected.

The fitting test with the phantom showed the shape retention and correct fit of the surgical guide on the reproduced phantom anatomy (Figure 7), as expected.

## 4. Discussion

The current study demonstrates: (1) the feasibility of printing a 3D surgical guide; (2) the fitting of the guide with a patient-specific heart model. Furthermore, a workflow of the 3D-printed surgical guide from the CT scan was described in order to introduce a medical device useful to improve CABG surgery and to also help as a training tool.

### 4.1. Feasibility of a 3D Surgical Guide Printing

3D models have been demonstrated to be useful in pre-surgical planning for the treatment of congenital heart disease [9,10]. Indeed, the latter 3D models are not aimed to help the clinician during surgery, but only in pre-procedural planning.

The model hereby described is an innovative 3D-printed surgical guide which might assist in optimizing the CABG surgery scenario, facilitating not only the recognition of stenosis, but also the selection of proper areas for bypass placement. This may lead to a decrease in procedural time.

To demonstrate the feasibility of a 3D surgical guide for bypass placement during CABG surgery, a 3D heart phantom was generated and printed. The physical surgical guide complied with the virtual model, according to its dimensions. The fitting test provided the correct positioning of the surgical guide on the heart phantom and demonstrated the effectiveness of the design choices. As expected, the surgical guide model was endowed with a thickness of 2.5 mm and exhibited the desired flexibility and stability, enough to remain fixed to the phantom. The inner surface, the face of the guide in contact with the epicardium, was smooth and suited, without excessive roughness. The outer surface was realized in glossy mode, obtaining a rigid external layer resistant to damage from medical instrumentation used during CABG surgery, such as a scalpel.

The increased thickness of the device in correspondence with the two reference points guaranteed the stability of the structure and the correct positioning of the guide on the heart template: the aortic ring was anchored to the aorta in order to guarantee the stability, as well as the apex.

The regions of interest, marked by surgical holes, allow for identification of the correct target for the bypass placement. The holes were deemed adequate for the intended use by physicians. Based on the feedback from an experienced cardiothoracic surgeon and an experienced cardiologist, the surgical guide was considered helpful and practical for clinical use.

### 4.2. Clinical Perspective

The implementation of a 3D surgical guide during a CABG procedure may be helpful for the cardiac surgeon to recognize a stenosis and to find the point for bypass placement. Furthermore, it can be used for research purposes and as a training tool. During CABG surgery, the presence of epicardial fat may increase the procedural difficulty and length of time. Indeed, fat can prevent the cardiac surgeon from obtaining a complete view of the pathological areas. The surgical guide developed, aimed at localizing the coronary tract appropriate for the bypass placement, could decrease procedure time and increase precision in lesion targeting. Moreover, the posterior wall of the heart is barely visible during CABG surgery, hindering the surgeon from locating the stenosis and the correct bypass position. This innovative tool does not aim to replace diagnostic CT images but integrates the information for a more efficient procedural workflow.

Therefore, the technology described in this study adheres to the principles of patient-oriented (or personalized) medicine where a 3D heart model can be individually tailored for each patient prior to coronary artery bypass graft surgery.

Further research addressing automatic segmentation software or faster printing techniques could represent a great improvement to the results found in this study.

Moreover, further clinical investigation to compare the conventional surgery with the surgical guide assisted coronary artery bypass graft surgery in terms of at least short-term efficiency, e.g., in-hospital outcomes, blood flow patterns, and quality of life (in particular, rehabilitation rates which are crucial for post-operative patients who have undergone such a surgical intervention) can be directed.

Another important future investigation includes the comparison of the efficiency of the designed technology for the freshmen and experienced surgeons. This holds another promise for the implementation of the automated workflow as it may particularly improve the results of coronary artery bypass graft surgery performed by junior surgeons.

Furthermore, the scope of this study was to demonstrate the feasibility of this surgical guide. Concerning clinical use, further clinical investigations and testing on the materials involved are eagerly awaited.

## Figures and Tables

**Figure 1 bioengineering-09-00179-f001:**
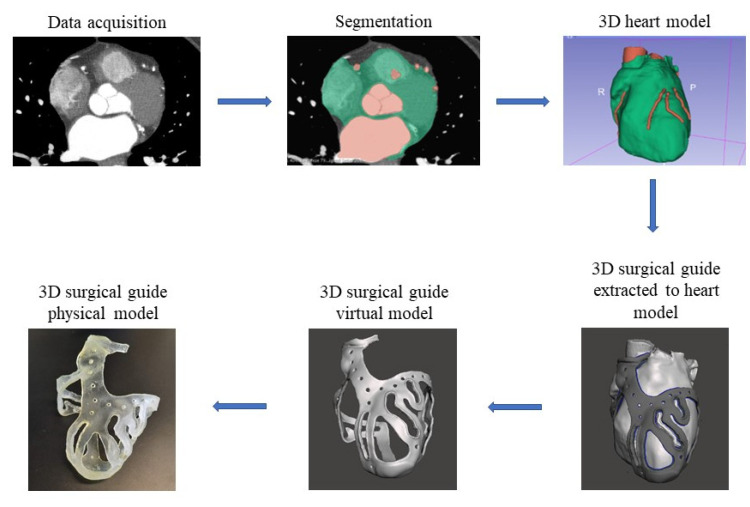
Workflow from the stack of medical images to the surgical guide.

**Figure 2 bioengineering-09-00179-f002:**
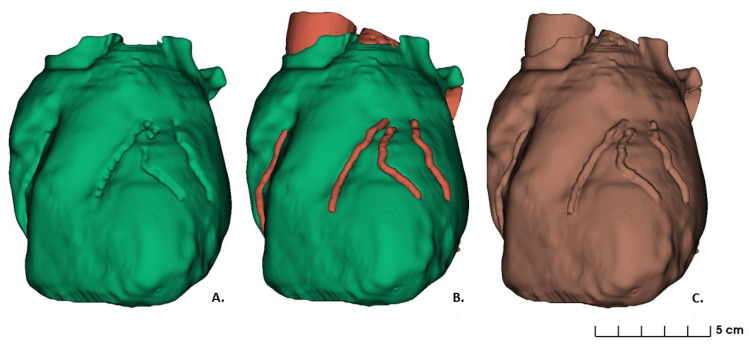
Heart model segmented: (**A**) First stage segmentation of whole heart without coronary artery tree; (**B**) Second stage segmentation as a combination of whole heart and coronary arteries; (**C**) Final segmentation stage.

**Figure 3 bioengineering-09-00179-f003:**
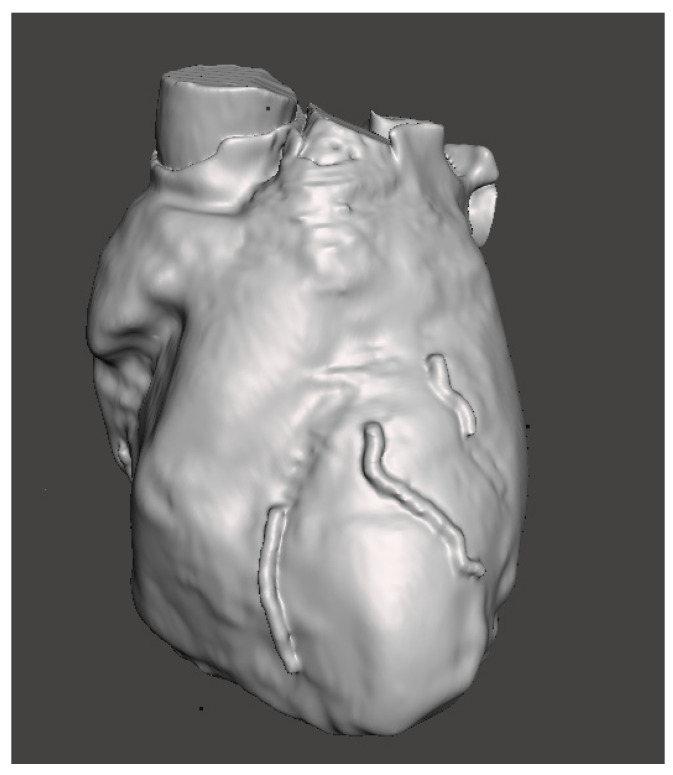
Heart model with selected region of interest.

**Figure 4 bioengineering-09-00179-f004:**
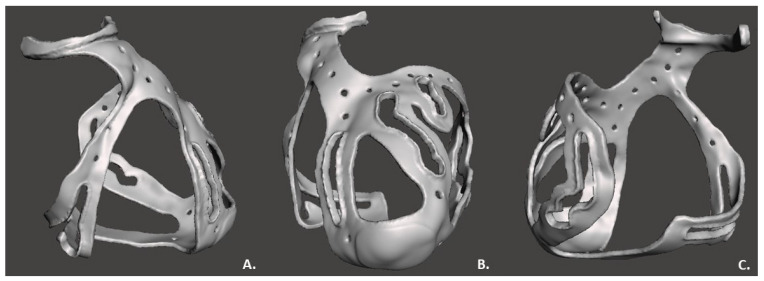
3D surgical guide in three views: (**A**) Lateral view; (**B**) Frontal view; (**C**) Posterior view.

**Figure 5 bioengineering-09-00179-f005:**
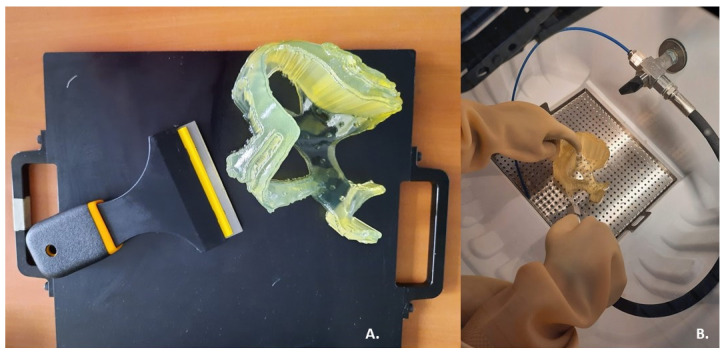
Post-printing steps: (**A**) Removal of the prototype from the 3D printer plate; (**B**) Cleaning of support material using a high-pressure water jet.

**Figure 6 bioengineering-09-00179-f006:**
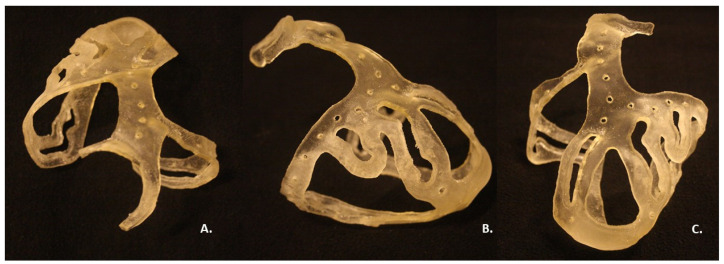
Printed and cleaned surgical guide: (**A**) Above view; (**B**) Left lateral view; (**C**) Frontal view.

**Figure 7 bioengineering-09-00179-f007:**
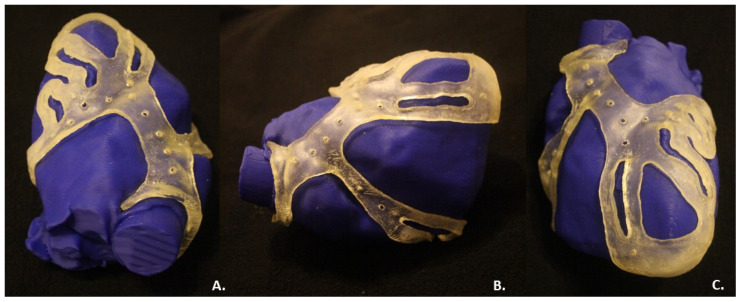
Fitting test of the surgical guide on the phantom heart: (**A**) Above view; (**B**) Right lateral view; (**C**) Frontal view.

## Data Availability

Not applicable.

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
