# Peer review of "3D Printed Surgical Guide for Coronary Artery Bypass Graft: Workflow from Computed Tomography to Prototype"

_bioengineering, 2022, doi:10.3390/bioengineering9050179_

Round 1

Reviewer 1 Report

Review of article: ‘3D Printed Surgical Guide for Coronary Artery Bypass Graft: Workflow from Computed Tomography to Prototype’ – Bioengeneering

The article by Cappello et al. presents a three-dimensional heart model aimed to help surgeons during coronary artery bypass (CABG) procedures.

     In performing CABG surgeons have so far relied basically on the information provided by coronary angiography and more recently by coronary angio-computed tomography. The present paper described another tool which has proved its feasibility in providing further information to the surgeon. The authors present a very sophisticated method to obtain a surgical prototype but from the picture included I do not really understand how these images could provide additional and even more useful information besides that already obtained by standard methods. It is also not clear whether this tool should be employed in the preoperative diagnosis or in the final decision-making in the operative room. In the second case I find it quite unpractical.

     Technological advancements are always important but they should not imply increased costs and should not make more cumbersome some already complex procedures. Concerning the specific tool described I would suggest to wait its clinical application to verify its usefulness besides the demonstrated feasibility.

Author Response

Reviewer #1:

The article by Cappello et al. presents a three-dimensional heart model aimed to help surgeons during coronary artery bypass (CABG) procedures.

In performing CABG surgeons have so far relied basically on the information provided by coronary angiography and more recently by coronary angio-computed tomography. The present paper described another tool which has proved its feasibility in providing further information to the surgeon. The authors present a very sophisticated method to obtain a surgical prototype but from the picture included I do not really understand how these images could provide additional and even more useful information besides that already obtained by standard methods. It is also not clear whether this tool should be employed in the preoperative diagnosis or in the final decision-making in the operative room. In the second case I find it quite unpractical.

A: Thank you for this comment; indeed, our tool is not intended to replace the standard images but to give information that are additive to the standard clinical images; in particular the tool is useful for the correct targeting of posterior wall of the heart and to help the surgeon in the visualization of the vessels embedded in the epicardial fat. As stated in the discussion, (page 9, line 267): “During CABG surgery the presence of the epicardial fat, may increase procedure challenge and timing; indeed, fat can prevent the cardiac surgeon from a complete view of the pathological areas.”. To provide further insights into the clinical use in the operative room, we added the following sentence in the discussion (page 9, line 271): “Moreover, the posterior wall of the heart is barely visible during CABG surgery, hindering the surgeon from finding the stenosis and the correct bypass position.”. Furthermore, based on the feedback of an experienced cardiothoracic surgeon (M.L.M.) and of an experienced cardiologist (C.d.A.) the surgical guide is helpful and practical for clinical use. We modified the discussion as follows (page 8, line 261): “Based on the feedback of an experienced cardiothoracic surgeon (M.L.M.) and of an experienced cardiologist (C.d.A.) the surgical guide was considered helpful and practical for clinical use.”

Technological advancements are always important but they should not imply increased costs and should not make more cumbersome some already complex procedures. Concerning the specific tool described I would suggest to wait its clinical application to verify its usefulness besides the demonstrated feasibility.

A: Thank you for this comment; the assessment of the clinical application was behind the scope of this study. We agree with the reviewer that further clinical investigation is eagerly awaited. It will be evaluated in further studies. We modified the discussion as follows (page 9, line 291): “Furthermore, the scope of this study was to demonstrate the feasibility of this surgical guide. Concerning the clinical use, further clinical investigation and testing on the materials involved is eagerly awaited.”

Reviewer 2 Report

Dear Authors,

Thank you for your interesting anlyisis.

Can you inform us in how pts you were implemented this 3D model?

Did your 3D model evaluated by cardiologist or cardiac surgeon? Their report was confirm your results?

Author Response

Reviewer #2:

Dear Authors,

Thank you for your interesting analysis.

Can you inform us in how pts you were implemented this 3D model?

A: Thank you for this comment; the patient-specific surgical guide was implemented based on the CT of one patient, to demonstrate the feasibility of the workflow. Future studies will focus on the standardization of the workflow, on a larger cohort.

Did your 3D model evaluated by cardiologist or cardiac surgeon? Their report was confirm your results?

A: Thank you for this comment; based on the feedback of an experienced cardiothoracic surgeon (M.L.M.) and of an experienced cardiologist (C.d.A.) the surgical guide is helpful and practical for clinical use. We modified the discussion as follows (page 8, line 261): “Based on the feedback of an experienced cardiothoracic surgeon (M.L.M.) and of an experienced cardiologist (C.d.A.) the surgical guide was considered helpful and practical for clinical use.”

Reviewer 3 Report

It was a nice study about the fabrication of a patient-specific 3D printed surgical guide from CT scan data for the aim of visualizing the region of interest for the bypass placement during the operation. Here are just two comments to improve the quality of the manuscript:

  • Please check the text again; there are some grammatical mistakes that should be corrected.
  • Please also compare the fabricated model with other 3D printed model in discussion part to high-light the benefits of your work against them.

Author Response

Reviewer #3:

It was a nice study about the fabrication of a patient-specific 3D printed surgical guide from CT scan data for the aim of visualizing the region of interest for the bypass placement during the operation. Here are just two comments to improve the quality of the manuscript:

Please check the text again; there are some grammatical mistakes that should be corrected.

A: Thank you for this comment. the text has been completely revised with the help of a native English speaker (Prof. Gian Battista Chierchia, co-author).

Please also compare the fabricated model with other 3D printed model in discussion part to high-light the benefits of your work against them.

A: Thank you for this comment that gave us the possibility to explain better how this innovative 3D tool differs from those studied in the recent years. We added the following sentence in the discussion (page 8, line 239): “Indeed, the latter 3D models are not aimed to help the clinician during surgery, but only in the pre-procedural planning. The model hereby described is an innovative 3D printed surgical guide which might assist in optimizing the CABG surgery scenario, facilitating not only the recognition of stenosis, but also the selection of proper areas for the bypass placement. This may lead to a decrease in procedural time.”

Reviewer 4 Report

Here, Cappello and colleagues presented a timely and important study describing an innovative workflow for the production of a patient-specific 3D printed surgical guide. The mentioned workflow included data acquisition using a multislice computed tomography, transfer of DICOM images into the 3D Slicer software, sequential reconstriction of whole heart surface reconstruction and aortic arch/coronary artery tree, superimposition of 2 reconstructions to reach a final image segmentation, transfer of a virtual 3D model into STL file for 3D printing, post-processing refinement in Meshmixer software, creation of a virtual surgical guide prototype, and printing of the polylactide phantom in a real scale to fit and to realise a surgical guide.

To me, the paper has no glaring omissions and deserves to be accepted for publication in Bioengineering as is. The technology described in the article might assist in optimising the coronary artery bypass graft surgery scenario before the intervention, facilitating recognition of stenosis and selection of the place for bypass placement. It is also an excellent training tool for medical students and surgical residents. Importantly, the tool does not replace CT images, rather integrating the 3D reconstruction information into more efficient procedural workflow. As stated by the authors, the study both highlights the technical opportunity for the virtual modeling and printing of a 3D surgical guide and shows successful fitting of such guide with a patient-specific heart model. The technology described in the paper therefore adheres to the principles of patient-oriented (or personalised) medicine where 3D heart model can be tailored for each patient individually before the coronary artery bypass graft surgery.

It would be productive if subsequent studies would compare conventional and virtually assisted coronary artery bypass graft surgery in terms of at least short-term efficiency, e.g. in-hospital outcomes, blood flow patterns, and quality of life (in particular rehabilitation rate which is crucially important for the patients who underwent such surgical intervention). There is no doubt that virtual design of the complex interventions such as coronary artery bypass graft surgery holds the future of precision medicine, yet the medical community is eager for the results of such comparative trials which might be organised similar to randomised clinical trials of novel drugs.

Another point for the future investigations includes the comparison of efficiency of the designed technology for the freshmen and experienced surgeons; one might suggest that it would be particularly efficient for the surgeons with < 5 years of experience. This holds another promise for the implementation of the automated workflow as it may particular improve the results of coronary artery bypass graft surgery performed by the young surgeons (who currently have lower success rates and this can be corrected!)

The paper might be featured on a journal cover as the Editor's choice.

Author Response

Reviewer #4:

Here, Cappello and colleagues presented a timely and important study describing an innovative workflow for the production of a patient-specific 3D printed surgical guide. The mentioned workflow included data acquisition using a multislice computed tomography, transfer of DICOM images into the 3D Slicer software, sequential reconstriction of whole heart surface reconstruction and aortic arch/coronary artery tree, superimposition of 2 reconstructions to reach a final image segmentation, transfer of a virtual 3D model into STL file for 3D printing, post-processing refinement in Meshmixer software, creation of a virtual surgical guide prototype, and printing of the polylactide phantom in a real scale to fit and to realise a surgical guide.

To me, the paper has no glaring omissions and deserves to be accepted for publication in Bioengineering as is. The technology described in the article might assist in optimising the coronary artery bypass graft surgery scenario before the intervention, facilitating recognition of stenosis and selection of the place for bypass placement. It is also an excellent training tool for medical students and surgical residents. Importantly, the tool does not replace CT images, rather integrating the 3D reconstruction information into more efficient procedural workflow. As stated by the authors, the study both highlights the technical opportunity for the virtual modeling and printing of a 3D surgical guide and shows successful fitting of such guide with a patient-specific heart model. The technology described in the paper therefore adheres (or personalised) medicine where 3D heart model can be tailored for each patient individually before the coronary artery bypass graft surgery.

A: Thank you for this comment that gave us the possibility to provide further insights into personalized medicine, towards the development of patient-specific tools. We added in the text these insights as follows (page 9, line 276): “The technology described in the study, therefore, adheres to the principles of patient-oriented (or personalized) medicine where 3D heart model can be tailored for each patient individually before the coronary artery bypass graft surgery.”

It would be productive if subsequent studies would compare conventional and virtually assisted coronary artery bypass graft surgery in terms of at least short-term efficiency, e.g. in-hospital outcomes, blood flow patterns, and quality of life (in particular rehabilitation rate which is crucially important for the patients who underwent such surgical intervention). There is no doubt that virtual design of the complex interventions such as coronary artery bypass graft surgery holds the future of precision medicine, yet the medical community is eager for the results of such comparative trials which might be organised similar to randomised clinical trials of novel drugs.

A: Thank you for this comment, we appreciate your suggestions for future research. We added your suggestion in the manuscript as follows (page 9, line 281): “Also, further clinical investigation to compare the conventional surgery with the surgical guide assisted coronary artery bypass graft surgery in terms of at least short-term efficiency, e.g. in-hospital outcomes, blood flow patterns, and quality of life (in particular rehabilitation rate which is crucially important for the patients who underwent such surgical intervention) can be directed.”

Another point for the future investigations includes the comparison of efficiency of the designed technology for the freshmen and experienced surgeons; one might suggest that it would be particularly efficient for the surgeons with < 5 years of experience. This holds another promise for the implementation of the automated workflow as it may particular improve the results of coronary artery bypass graft surgery performed by the young surgeons (who currently have lower success rates and this can be corrected!)

A: Thanks for this comment. We agree with the reviewer and we add these suggestions as follows (page 9, line 286): “Another important future investigation includes the comparison of the efficiency of the designed technology for the freshmen and experienced surgeons. This holds another promise for the implementation of the automated workflow as it may particularly improve the results of coronary artery bypass graft surgery performed by the young surgeons.”

The paper might be featured on a journal cover as the Editor's choice.

A: Thanks for this comment. We appreciate the interest in our research and we hope this paper can be an important scientific contribution to this journal.

Round 2

Reviewer 1 Report

I think that the paper can  now be accepted